# Cannabis Seedlings Inherit Seed-Borne Bioactive and Anti-Fungal Endophytic Bacilli

**DOI:** 10.3390/plants11162127

**Published:** 2022-08-15

**Authors:** Christopher R. Dumigan, Michael K. Deyholos

**Affiliations:** Irving K. Barber Faculty of Science, University of British Columbia Okanagan, Kelowna, BC V1V 1V7, Canada

**Keywords:** endophyte, biocontrol, bacillus, fusarium, alternaria, aspergillus, penicillium, damping-off, biological mediated management, pathogens

## Abstract

Throughout the hundreds of millions of years of co-evolution, plants and microorganisms have established intricate symbiotic and pathogenic relationships. Microbial communities associated with plants are in constant flux and can ultimately determine whether a plant will successfully reproduce or be destroyed by their environment. Inheritance of beneficial microorganisms is an adaptation plants can use to protect germinating seeds against biotic and abiotic stresses as seedlings develop. The interest in *Cannabis* as a modern crop requires research into effective biocontrol of common fungal pathogens, an area that has seen little research. This study examines the seed-borne endophytes present across 15 accessions of *Cannabis* grown to seed across Western Canada. Both hemp and marijuana seedlings inherited a closely related group of bioactive endophytic *Bacilli.* All *Cannabis* accessions possessed seed-inherited *Paenibacillus mobilis* with the capacity to solubilize mineral phosphate. Additionally, seeds were found to carry genera of fungal isolates known to be *Cannabis* pathogens and post-harvest molds: *Alternaria*, *Penicillium*, *Cladosporium*, *Chaetomium*, *Aspergillus*, *Rhizopus*, and *Fusarium*. Thirteen seed-borne endophytes showed antibiotic activity against *Alternaria, Aspergillus, Penicillium,* and *Fusarium*. This study suggests both fungal pathogens and bacterial endophytes that antagonize them are vectored across generations in *Cannabis* as they compete over this shared niche.

## 1. Introduction

Plants can adapt to their environment by hosting beneficial bacteria and fungi that confer selective advantages to their host in stressed conditions [1]. These microorganisms can contribute to overall plant health by assisting in nutrient acquisition, secreting plant-growth-promoting phytohormones, eliciting host immune responses, or antagonizing pathogens to prevent disease [1,2,3,4]. Microorganisms that exist within plants are known as “endophytes”, and current research has elucidated how important these microorganisms are with respect to plant health [5,6]. The internal spaces of plants are unsurprisingly ideal habitats for many microorganisms: with a rich supply of carbon and other nutrients, as well as protection from harsh environments, plants host vast communities of microorganisms that can have a beneficial or pathogenic effect on their host [1,2,3,4,7].

In addition to the microorganisms obtained from the environment, plants inherit a founding microbiome via seed-to-seedling vertical transmission [8]. Developments in our understanding of the seed microbiome and its role in plant ecology have been slow in comparison to that of the interaction with soil, the rhizosphere, and the plant microbiome as a whole [9]. Seed-borne microorganisms are of particular interest as they act as the starting point for the assemblage of new microbial communities for germinated seedlings during this critical stage in the plant life cycle [10].

Research suggests that the microbial communities associated with seeds planted in soil originate primarily from the mother plant rather than from their soil environment [11,12]. This seed microbiome is an important source of inoculum for developing seedlings, and plants appear to place strong selective forces on seed-borne microorganisms over soil-borne microorganisms during germination by shaping what species proliferate and colonize emerging seedlings [7]. Microorganisms that have effectively colonized plant tissues have the potential to exert a “barrier effect” against other microorganisms, preventing their colonization [13,14]. This potentially explains why seed-borne microorganisms, especially endophytes, are not simply overtaken by the ocean of soil-borne microorganisms and can persist across plant generations [15].

Soil is a well-established driver of the composition of plant-associated microbial communities. However, understanding of the role for seed-borne microorganisms in the plant life cycle has grown substantially throughout the past decade [9]. While research has suggested that the bacterial communities of rhizospheres are primarily influenced by the surrounding soil [16], there is also evidence that microbial inheritance plays an important role in forming early rhizosphere communities. One such study found that while soil type was the main driver of bacterial group structure in the rhizosphere, the most common bacterial cells (Proteobacteria and Bacteroidetes) in juvenile maize rhizospheres across three soil types appeared to be seed-derived. [17]. These results are further supported by a recent study showing that soil adds microbial diversity to early plant rhizospheres, while seeds contribute the most abundant bacteria found during the early stages of a plant’s life cycle [12].

This phenomenon of microbial inheritance appears to play a major role in determining microbial communities in plant endospheres [18]. Bacterial endophytes of maize have been suggested to be primarily seed-borne rather than originating from their soil environment [18]. Given that this phenomenon of microbial inheritance appears to play a major role in determining microbial communities in plant endospheres, it is interesting to speculate about alternative systems outside of genetics by which selective advantages are inherited from parental plants. There is potentially selection pressure for microorganisms to be maintained and transmitted to offspring plants to protect them against biotic and abiotic stresses. Researchers have suggested that certain microorganisms can protect developing seedlings from pathogens in their new environment by altering seedling redox status [19]. Furthermore, a diversity of seed endophytes isolated from cucurbit species showed antibiotic activity against common greenhouse pathogens such as powdery mildew and *Pythium* [6,20]. It is clear that plants and microorganisms have co-evolved over hundreds of millions of years and that plants have selected for symbionts that are vertically transmitted across generations to provide a selective advantage to germinating seedlings [9,12].

The emerging global *Cannabis* market is predicted to reach 30 billion USD by 2025 [21] and the demands of growing a new crop on this scale will present an array of problems that require creative and sustainable solutions. Given the nature of the market, little research has been done on *Cannabis* cultivation relative to other crops, and so there are many opportunities to answer important questions that will benefit growers in Canada and around the world. *Cannabis* growers are challenged by an array of fungal pathogens with the potential to destroy entire harvests. Furthermore, synthetic fungicides are forbidden for both recreational and medicinal *Cannabis* in Canada and California (https://bit.ly/2BMfIYt (accessed on 1 March 2021), https://bit.ly/2OkvT4T (accessed on 1 March 2021). Consumer demand for safe, organically grown *Cannabis* products means that growers need effective biocontrol agents for fungal pathogens, an area that has seen little research in *Cannabis.*

Recent research on pathogens of *Cannabis* has documented what growers have struggled with long before the recent legalization of *Cannabis* in Canada: roots and stems of *Cannabis* plants are commonly infected by species of *Fusarium* and *Pythium* [22,23,24], both of which cause wilting and stem rot—symptoms colloquially known as “damping off”. The dominant foliar pathogen associated with *Cannabis* tissue is *Golovinomyces cichoracearum,* which causes powdery mildew [23]. Although this pathogen can also infect stems and flowers, it most commonly presents itself as white powdery mycelial growth on leaves [25]. The revived interest in *Cannabis* is primarily driven by the secondary metabolites produced in flowering tissue. *Cannabis* female inflorescences (commonly known as “buds”) are infected by several pre- and post-harvest pathogens with the potential to make the final product unusable. *Botrytis*, *Penicillium*, and *Fusarium* are causes of “bud rot” in both pre- and post-harvest tissues [25,26]. Species of well-studied plant pathogens, *Cladosporium*, *Alternaria*, and *Aspergillus,* have also been isolated from flower tissue grown indoors and in greenhouses [23,25]. These genera of flower-infecting fungi are known to have toxigenic species that produce mycotoxins that cause illness, or, at the very least, lead to quality control failures [27,28].

In general, microbial inheritance via seed-to-seedling vertical transmission occurs through microbial colonization of external and internal seed tissue in developing flowers [29,30]. In fact, microscopic analysis shows that the inner surface of flower petals is densely populated by a diversity of bacteria, including species that show biocontrol activity [31]. These species are in close contact on the surface of the seed coat and have the potential to be passed down to developing seedlings as they germinate [9]. Given that a diversity of fungal pathogens readily colonize *Cannabis* flowers both pre-and post-harvest, we hypothesize that *Cannabis* seeds vector opportunistic fungal pathogens as well as bacterial symbionts that protect them against biotic and abiotic stresses as they germinate into seedlings. Throughout our report, the mature *Cannabis* fruits (achenes) are referred to as “seed” for simplicity and because they are commonly referred to as such in the hemp and marijuana industry. This whole *Cannabis* fruit is what is typically sown in fields or naturally shed from *Cannabis* plants when ripe [32,33] and so is analogous to seeds of other crops in the context of seed-borne microorganisms [12].

The objective of this study is to (1) survey seed-borne endophytes found inside juvenile *Cannabis* seedlings. A selection of 15 accessions of hemp and marijuana types were grown to maturity in different locations across Western Canada before seeds were germinated in the absence of environmental microorganisms. A total of 136 bacterial isolates were cultured, identified, and (2) phenotyped for their capacity to solubilize mineral phosphate, secrete indole-3-acetic acid (IAA), and hydrolyze pectin. A selection of unique isolates from each accession (78 endophytes) were then tested for their ability to (3) antagonize seed-borne fungi that had been isolated from germinating *Cannabis* seeds. We demonstrated that the *Cannabis* accessions in this study inherited a conserved group of Bacilli with bioactivites that included suppression of fungal growth. This study suggests co-evolution between *Cannabis* and vertically transmitted, endophytic Bacilli.

## 2. Results

### 2.1. Cannabis Seedlings Inherit Seed-Borne Endophytic Bacilli

We germinated seeds from 15 different accessions of *Cannabis* including both hemp and marijuana types (Figure 1a). These seeds were germinated in sterile Petri plates, and the culturable endophytes within the germinated seedlings were identified (Figure 1b). Because the culture environment had been sterilized, the endophytes present inside seedling tissue must have been acquired from the seed surface and seed endosphere [12]. Prior to detection of endophytes, all seedlings were tested for surface sterility by plating a final wash onto nutrient agar; no growth was observed on any plate, which confirms that seedling surfaces were sterile. A total of 136 bacterial isolates were obtained from the seedling endosphere and were assigned to taxa using 16S sequencing. Although the seeds came from four different locations and growing conditions in Canada, we observed unexpected conservation of some of the culturable, seed-borne seedling endophytes (Table 1). Moreover, seeding endospheres of these 15 *Cannabis* accessions appeared to be dominated by endospore-forming bacteria in the class Bacilli. Notably, *Paenibacillus mobilis* isolates were cultured from endospheres of 100% (15/15) of accessions, despite the different conditions in which their mother plants were grown (Figure 1a). The well-studied and agriculturally important soil bacterium, *Bacillus subtills,* was found in the majority (8/15) of *Cannabis* accessions, and was present in both hemp and marijuana.

Seed from field-grown flax, *Linum usitatissimum* var.Bethune, was included in this experiment as an out-group for comparison and as a control for contamination. Only one culturable seed-borne endophyte, *Bacillus megaterium,* was isolated from flax seedlings (Table 1). This bacterium was found in 47% of the *Cannabis* genotypes (7/15). Seed-borne *Paenibacillus mobilis* was not isolated from any flax endospheres in our experiments.

Phylogenetic analysis of the 16S gene from the 136 seed-borne *Cannabis* bacterial endophytes further demonstrated that a large number of the culturable bacterial taxa that colonize *Cannabis* seedling endospheres are well-conserved across seed accessions (Figure 2). Endophytes belonging to the genera *Bacillus* (28.26%) and *Paenibacillus* (67.39%) made up the majority of isolates. *Bacillus* endophytes were isolated from 80% of *Cannabis* genotypes, while endophytes belonging to the genus *Paenibacillus* were isolated from all 15 (100%) *Cannabis* accessions. In the absence of environmental microorganisms, the *Cannabis* genotypes in this study appeared to inherit a group of closely related endophytes from parental seed. Details on all isolates obtained from the seed-borne endospheres can be found in Appendix A.

Hemp variety X59 was represented in our analysis by two samples from two different generations, grown under different conditions. The first generation (X59 Gen-1) was produced by field-grown plants in Vegreville, Alberta. Some of these seeds were subsequently sown in soilless commercial potting mix and grown to maturity indoors in Kelowna, British Columbia, and the seeds produced by these plants were labeled X59 Gen-2. Analysis of endophyte 16S amplicons showed that the strains of *Bacillus megaterium* and *Paenibacillus mobilis* isolated from seedling endospheres across these generations were identical despite growing in vastly different environments (Appendix A).

### 2.2. Endophyte Phosphate Solubilization

Plants have been known to harbor root endophytes that secrete organic acids to solubilize rock phosphate in the surrounding environment. Seed-borne *Cannabis* endophyte isolates were assessed for their ability to solubilize mineral phosphate by growing on tricalcium phosphate media and creating clear halos around the culture—interestingly, phosphate-solubilizing bacteria were isolated from the endosphere of all 15 *Cannabis* accessions (100%). In total, 10 different bacterial species were found to solubilize phosphate. Ubiquitous across all *Cannabis* seedings in this study was phosphate-solubilizing *Paenibacillus mobilis* (Figure 3).

### 2.3. Endophyte Indole-3-Acetic Acid Activity

Plant-associated bacteria have been known to secrete phytohormones that interact with their host plant with the potential to modulate plant growth. Microbial production of the auxin, indole-3-acetic acid (IAA), can promote root growth and colonization, and circumvent plant immune responses [34,35]. *Cannabis* endophytes were assayed for in vitro IAA production and secretion using a colorimetric assay [36]. IAA production was detected in six different bacterial species isolated from 53.3% (8/15) of *Cannabis* accessions in this study (Figure 3).

### 2.4. Endophyte Pectinase Activity

Pectin is a structural heteropolysaccharide that is a major component of plant cell walls. Pectinase activity of seed-borne *Cannabis* endophytes was assessed by growing on agar supplemented with citrus pectin and visualizing halos by staining with Gram’s Iodine. Pectinase activity was found in four different bacterial species across five *Cannabis* accessions (33%) (Figure 3). Pectinase activity was limited to endophytes of hemp, as no marijuana accessions hosted species with this phenotype.

### 2.5. Seed-Borne Fungi

Seedling endospheres are expected to harbor fungi [10]. However, following seedling surface sterilization, only a single accession of either *Cannabis* or flax produced culturable fungus on PDA with tetracycline. This isolate was obtained from the God’s Bud accession and was identified as PM17- *Penicillium* (Appendix A). In contrast, bacterial endophytes were cultured from seedling endospheres of every accession, totaling 136 bacterial isolates.

Seeds are a well-known vector of both fungal symbionts and pathogens in many plant species [28,37,38]. The absence of fungi from surface-sterilized seedlings does not imply that they are absent from seeds and seedling surfaces. Therefore, *Cannabis* accessions were examined without seedling surface sterilization where seeds were instead directly germinated on PDA with tetracycline to assess germination and presence of seed-associated fungi. In this experiment, “VIR577” was substituted for “God’s Pink” due to availability of seeds.

Fungal isolates were cultured from seeds/seedlings of 73.3% (11/15) of *Cannabis* accessions. From each accession, morphologically unique colonies were subcultured and identified by ITS region sequencing. A total of 24 seed-associated fungal isolates were found from this experiment (Figure 4a). Of interest, 83.3% (20/24) of these isolates were identified as genera of known *Cannabis* pathogens and post-harvest molds: *Alternaria, Penicillium, Cladosporium, Chaetomium, Aspergillus, Rhizopus,* and *Fusarium.* The germination rates and number of seeds engulfed by mycelia (disease pressure) in this experiment can be seen in Figure 4b. Notably, there were large fluctuations in the germination rates of the various *Cannabis* accessions as well as three cultivars with 100% germination and 0% of seeds with any fungal disease pressure: Afghani, Alyssa, and White Widow.

### 2.6. Endophyte Anti-Fungal Activity

A total of 78 seed-borne bacterial endophytes were selected based on unique taxonomies from each accession of *Cannabis* in this study. These isolates were tested for their ability to antagonize growth of four different seed-borne fungi belonging to genera of known *Cannabis* pathogens: *Alternaria* PM35, *Aspergillus* PM40, *Penicillium* PM52, and *Fusarium* PM60 (Figure 5). Thirteen bacterial isolates covering four different species were found to have anti-fungal properties and were capable of creating zones of inhibition when co-cultured on PDA with these fungi. Anti-fungal activity was found in bacterial isolates from 60% (9/15) of the *Cannabis* accessions examined in this study. A summary of these data can be seen in Table 2.

*Alternaria* PM35 (Figure 5a) was the fungal isolate that was antagonized by the largest number of bacterial isolates tested, with 13 bacterial isolates inhibiting *Alternaria* PM35. *Bacillus subtills* OKM128 was the strongest antagonist and created an average zone of inhibition equal to 38.6 ± 11.02 mm in diameter after 7 days of co-culturing. In contrast, *Aspergillus* isolate PM40 (Figure 5b) was inhibited by only four of the bacterial isolates tested, making it the fungus that showed the highest level of resistance in this study (Table 2). Anti-*Aspergillus* isolates were limited to the species *Paenibacillus polymyxa* and *Bacillus velezensis;* the largest average zone of inhibition was 11.0 ± 1.0 mm from *Bacillus velezensis* OKM141. Five isolates comprising three different species were able to antagonize *Penicillium* PM52: *Bacillus subtilis, Paenibacillus polymyxa,* and *Bacillus velezensis. Bacillus subtills* OKM128 produced, again, the largest zone of inhibition with an average diameter of 18.33 ± 2.5 mm. Zones of inhibition against *Fusarium* PM60 were produced by 12 isolates (three different species), the greatest of which was an average of 17.67 ± 2.3 mm by *Paenibacillus polymyxa* OKM170. All negative controls (uninoculated TSB liquid medium) showed no zones of inhibition, with fungal hyphae growing over holes in agar. ANOVA and Tukey’s HSD test for significance were conducted across bacterial isolates (including negative control) for antagonism of each fungus. Letter values (Figure 5) were assigned to means to distinguish groups significantly different from one another (α = 0.05).

## 3. Discussion

Plant microbiome research has been heavily focused on the role of soil on microbial community structure. Until recently, the dominance of seed-borne microorganisms in early community establishment was poorly understood. New research demonstrates that seed-borne microorganisms are the primary colonizers of juvenile angiosperms, and soil microorganisms only colonize young plants efficiently when the seed microbiome is disrupted by surface sterilization [12,39]. Importantly, most previous research on the contributions that plant genotype, seed microorganisms, and soil communities have on the plant microbiome has used surface-sterilized seeds, disrupting spermosphere/seed-coat microorganisms that play an important role in vertical transmission [40]. As has been done in more current studies, we deliberately did not surface sterilize *Cannabis* seeds, prior to seedling surface sterilization, to avoid artificially disrupting seed-surface microorganisms that may be heritable [9,12,18,39]. Despite the diverse geographic origins of seed production, *Cannabis* seedlings in this study, when germinated in the absence of environmental microbes, inherited a closely related group of endophytic Bacilli with a variety of interesting phenotypes that may benefit seeds as they germinate in a new environment (Table 1, Figure 3).

The phenomenon of microbial inheritance is well-studied in many plant species [10]. Our report documents this process in *Cannabis* and identifies some of the relevant functions of seed-borne seedling endophytes. To further support the previous observation, a second generation of the hemp cultivar X59 was grown to maturity to produce seed indoors in Sunshine soil mix (Sun Gro Horticulture, Agawam, MA, USA) at the University of British Columbia, Okanagan. This environment differs greatly from the outdoor fields, where X59 generation-1 was grown to maturity to produce seed at InnoTech Alberta (Figure 1). The clear preservation of two identical species of seed-borne endophytes across this generation suggests that X59 (and *Cannabis* in general) selects for certain microorganisms to be maintained in its microbiome and passed down subsequent plant generations (Appendix A). Other plants such as rice and corn have conserved and vertically transmitted core microbial groups that appear to be maintained across generations from ancestral relatives to modern cultivars [41,42]. Future studies should investigate what core microbial species are found in *Cannabis* seeds throughout its evolution, domestication from the Himalayan foothills of central Asia, and thousands of years of cultivation [43,44]. Such a study would rely on improved phylogenetic classification of ancestral, landrace, and modern *Cannabis* accessions grown today, an area that is currently being researched [45].

All 15 accessions of *Cannabis*, including hemp and marijuana, inherited seed-borne *Paenibacillus mobilis* with the capacity to solubilize mineral rock phosphate. The out-group flax cultivar, Bethune-0, did not harbor this species of phosphate-solubilizing *Paenibacillus mobilis* (Figure 3). Phosphate is a vital macronutrient and plants have been documented to host endophytic bacteria in root zones that secrete organic acids that solubilize mineral phosphate, increasing bioavailability, and promoting root growth [2,46,47]. Of note, an endophyte was isolated from seeds of a wild maize relative growing in a volcanic swamp with high levels of insoluble mineral phosphate, and was likely selected to be vertically transmitted across plant generations to assist seedlings in this harsh environment [2]. The observation that all accessions of *Cannabis* hosted seed-vectored *Paenibacillus mobilis* with the potential to convert tricalcium phosphate to a more plant-bioavailable form leads to speculation about what selection pressure nature has exerted on *Cannabis* throughout its evolution and domestication. Indoor and field-grown hemp and marijuana hosted this endophyte. Phosphorous and nitrogen have been documented to be the most important macronutrients contributing to *Cannabis* flower yield [48], and phosphorous nutrition can have an effect on cannabinoid profile [49]; however, research in this area is still developing.

Seed germination and seedling establishment is sensitive to oxidative imbalances. Reactive oxygen species (ROS) act both as an elicitor to release seed dormancy, a defense mechanism against disease, and toxic metabolites when accumulated in plant cells [50,51]. These conflicting effects create an “oxidative window for germination” that is tightly controlled by auxin hormone signaling pathways such as IAA to promote healthy germination [52,53,54]. IAA-secreting bacterial endophytes were found in 53.3% of *Cannabis* accessions examined in this study (Figure 3). Seed-borne endophytic bacteria that produce IAA have previously been documented to alter seedling development in rice [55]. This same study showed that when these seed-associated bacteria were removed, there were lower levels of ROS as well as a detrimental effect on root hair development and seedling size compared to rice with an intact seed-borne microbiome. It is speculated that the increased ROS production from seed-borne endophytes may also serve to protect seedlings against invading pathogens [55]. IAA-producing endophytes have been previously isolated from industrial hemp cultivars [56]. It is unsurprising that the *Cannabis* accessions examined here, similar to other plants, inherit bacteria that secrete IAA and likely contribute to seedling germination and establishment. Further testing of these endophytes and their effect on seed germination and establishment should be examined.

Hydrolysis of plant cell walls by enzymes such as pectinase, xylanase, and cellulase is thought to be a mechanism by which some endophytes gain entry into plant tissues [57]. A small number of bacteria in this study were able to grow on and hydrolyze citrus pectin (Figure 4). Further testing of this library of endophytes should examine other cellulolytic enzyme activity, as well as other sources of pectin. It should be noted that citrus pectin has been documented to have powerful antibiotic activity against both gram-negative and gram-positive bacteria [58]. There appeared to be growth inhibition of many endophytes in the pectinase assay which would lead them to score negative; these endophyte species may well be able to hydrolyze other forms of pectin.

The role of seeds in dispersal of fungal pathogens in *Cannabis* has not seen extensive research [59]; however, seed-borne fungal diseases are well-documented in established crops [28,37,38,60]. This study found that 73.3% (11/15) of *Cannabis* accessions hosted at least one seed-borne fungal isolate (Figure 4). The observation that 83.3% (20/24) of these isolates were identified as genera of known *Cannabis* pathogens and post-harvest molds: *Alternaria, Penicillium, Cladosporium, Chaetomium, Aspergillus, Rhizopus,* and *Fusarium,* is unlikely a coincidence. Rather, *Cannabis* seeds likely act as a vector for fungal pathogens to spread and proliferate. Of particular interest is the observation that *Penicillium, Fusarium, Cladosporium*, *Alternaria*, and *Aspergillus* have been previously identified infecting *Cannabis* flowers [23,25,26]. Isolates belonging to these pathogenic genera account for 70.8% (17/24) of total isolates. Seed-bone *Alternaria* has also recently been documented as a pathogen of mature *Cannabis* [61]. Taken together, this suggests that *Cannabis* flowers act as an infection point for opportunistic fungi to be vectored across plant generations via seed transmission. Observationally, the fungi isolated here from germinating seeds appeared to colonize and destroy seed and seedling tissue (Appendix A); however, further testing must be done to investigate their potentially pathogenic behavior to mature *Cannabis* plants.

Given the evidence that *Cannabis* seeds vector both a conserved group of endophytic Bacilli as well as fungi belonging to potentially pathogenic genera, there is likely competition and co-evolution for the shared niche of germinating *Cannabis* seeds, as well as selection pressure on *Cannabis* to maintain beneficial microorganisms with biocontrol in their microbiome. Members of class Bacilli produce a variety of antimicrobial secondary metabolites [62,63]. The genera *Bacillus* and *Paenibacillus* are currently the best characterized for producing lipopeptides and polyketides that have immense commercial value in agriculture for their capacity to antagonize pathogenic fungi and suppress crop disease. A total 60% (9/15) of *Cannabis* accessions in this study hosted seed-borne endophytic Bacilli that inhibited growth of seed-associated fungi belonging to plant pathogenic genera (Figure 5, Table 2). Four fungal isolates were selected for this assay based on their taxonomy belonging to the genera *Alternaria*, *Aspergillus*, *Penicillium*, and *Fusarium.* All of these are known plant pathogens and have been documented to infect *Cannabis* flowers and cause disease in *Cannabis* plants [23,25,26,61]. Of the 78 selected endophytes isolated from the 15 *Cannabis* accessions, 13 isolates showed antagonism against seed-associated *Cannabis* fungi (Figure 5, Table 2). Four endophytes showed antagonism against all fungi assayed: *Bacillus velezensis* OKM141, *Paenibacillus polyxma* OKM147, OKM155, and OKM170. These endophytes will be further studied for their ability to control fungal pathogens in mature *Cannabis* plants, as well as on seed. Given the observation of highly variable rates of germination, it is interesting to speculate about the contribution of these various fungi to the diminished germination seen in seeds of certain *Cannabis* accessions (Figure 4b). Seed-borne *Fusarium*, *Penicillium*, *Rhizopus*, and *Aspergillus* are associated with decreased seed germination in rice [28]. Future testing of seed-coating anti-fungal bacterial isolates on low-germination seed will be done to investigate antagonism of seed-associated fungal pathogens, and the resulting effect on germination. Additionally, these anti-fungal bacteria will be tested for their capacity to protect seeds by flower and foliar applications and establishment in the subsequent plant generation as seedling endophytes.

A variety of commercial products exist that use beneficial microorganisms to combat plant pathogens. Serenade^®^ was developed by AgriQuest.Inc (now owned by Bayer) and contains a species of *Bacillus subtills* that produces a variety of antimicrobial lipopeptides. Serenade^®^ is effective at controlling a variety of pathogens including: fungi such as botrytis [64], powdery mildew [65], yellow rust [66] and early blight [67]; and protists such as clubroot [68] and the oomycetes *Pythium* and *Rhizoctonia* [69]. Researchers have begun to investigate the potential of *Bacillus* species to control fungal pathogens and promote plant growth in *Cannabis* with promising results [70,71,72]. What makes *Bacillus* and *Paenibacillus* particularly attractive to commercial farming is their capacity to form desiccation-resistant endospores, allowing formulation and survival as seed coats [62]. The seed-borne endophyte library generated in this study is dominated by endospore-forming bacteria and so may possess strains with the potential to be useful as seed-coating biologicals in field-grown hemp. Furthermore, drug-type *Cannabis* (marijuana) is typically clonally propagated to mitigate variation in phenotypes observed when plants are grown from seed [73,74,75,76]. The results presented here lead to speculations about the different microbiomes of clonally propagated and seed-grown *Cannabis* plants. Whether these seed-borne anti-fungal seedling endophytes are absent in clonally propagated *Cannabis* plants would be of interest to growers of this horticultural crop with limited tools for control of fungal pathogens.

In conclusion, this study demonstrates that *Cannabis* seeds inherit bioactive and anti-fungal Bacilli that colonize seedling endospheres. Additionally, *Cannabis* seeds appear to be associated with fungi belonging to genera of known floral pathogens, suggesting that fungi may opportunistically use flowers as a transmission route to infect germinating *Cannabis* seedlings. The production of antimicrobial secondary metabolites that antagonize seed-associated fungi by these bacterial endophytes suggests competition for the shared niche of germinating *Cannabis* seeds. Taken together, this paper suggests co-evolution and potential symbiosis between *Cannabis* and seed-borne Bacilli endophytes.

## 4. Materials and Methods

### 4.1. Source of Seed

Bethune-0 is from Kernen Farm in Saskatoon. Altair, Alyssa, CAN3797, FIN34, Grandi, Katani, LKCSD, VIR577, and X59 generation-1 are from InnoTech Alberta. Canda is from Triple S Seeds LTD in Grandview, Manitoba. Afghani, BC Big Bud, God’s Bud, God’s Pink, and White Widow are from a commercial supplier in Qualicum Beach, British Columbia. X59 generation-2 was grown to produce seed at UBCO Kelowna, BC, Canada.

### 4.2. Seed-Borne Cannabis Seedling Endophyte Isolation

An overview of the isolation design can be seen in Figure 1b. Seedling surface sterilization was based on established methods for seeds with modification [42]. Five seeds of each *Cannabis* accession were germinated in sterile environments of Petri plates on autoclaved wet paper towel and placed in the dark for 5 days at 20 °C. Once germinated, seedlings were cleaned by washing with filter-sterilized 0.1% triton X for 10 min. Samples were then surface-sterilized by washing with 3% sodium hypochlorite for 20 min. The samples were then drained and rinsed with autoclaved, distilled water, then washed in 70% ethanol for 10 min. The ethanol was removed, and samples were rinsed 5 times with autoclaved, distilled water. To check for surface sterility, 150 μL of the final wash was spread on R2A and PDA, and one piece of tissue per treatment was transiently placed on sterile agar plates, which were all incubated for 10 days at 25 °C. Surface-sterilized seedlings were ground in 2 mL sterile phosphate buffer using a sterile mortar and pestle. An amount of 150 μL of this solution was spread on plates: R2A + 20 μg/mL nystatin, PDA + 20 μg/mL tetracycline, and TSA+ 20 μg/mL nystatin and incubated at 25 °C. Morphologically unique colonies from each Cannabis accession were isolated and subcultured from each media type. Isolates were grown in R2B for 3 days at 25 °C and stored in 20% glycerol at −20 °C.

### 4.3. Taxonomic Identification by 16S Gene Sequencing

Colony PCR of bacterial isolates was performed on 1 μL of R2B culture using primers 341F and 785R. PCR was conducted with the following settings: initial denaturation at 95 °C for 5 min, followed by 45 cycles of denaturation at 95 °C for 30 s, annealing at 55 °C for 30 s and extension at 72 °C for 1 min; a final extension phase was performed at 72 °C for 10 min. Amplicons were run on a 1% TAE gel to confirm successful amplification prior to sending to BioBasic (Markham) for Sangar Sequencing. Forward and reverse sequences were aligned and consensus sequences were identified using BLAST against the 16S ribosomal RNA sequences database. Highest scoring alignments were assigned as isolate taxonomy. Sequences were submitted to Genbank under accession numbers OP087663-OP087800.

### 4.4. Phosphate Solubilization Assay

Solubilization of mineral phosphate activity was tested by growing isolates on tricalcium phosphate media (10 g/L glucose, 2 g/L NH_4_CL, 0.41 g/L MgSO_4_, 0.295 g/L NaCl, 0.003 FeCl_3_, 0.7 g/L Ca_3_HPO_4_, 12 g/L Agar) and identifying halos around colonies after 3 days of growth at 25 °C [42,77]. Glycerol stocks were used to inoculate R2B medium which was grown for 2 days shaking at 20 °C. Cultures were inoculated on tricalcium phosphate media using a 96-pin replicator; this assay was done in triplicate.

### 4.5. Pectinase Activity

Pectinase activity was tested by growing isolates on R2A media supplemented with 0.2% citrus pectin (#J61021, Alfa Aesar) and 0.1% triton X-100. Glycerol stocks were used to inoculate R2B medium, which was grown for 2 days shaking at 20 °C prior to inoculating pectin media in triplicate using a 96-pin replicator. Pectin plates were placed in the dark at 25 °C for 3 days followed by flooding with Gram’s Iodine, where isolates displaying pectinase activity were surrounded by clear halos [42].

### 4.6. IAA Secretion

Indole-3-acetic acid secretion was tested by growing isolates on R2A media supplemented with 5 mM L-tryptophan. Glycerol stocks were used to inoculate R2B media, which was grown for 2 days shaking at 20 °C prior to inoculating tryptophan-supplemented R2A in triplicate using a 96-pin replicator, which was then grown for 2 days at 25 °C. Plates were then covered with a nitrocellulose membrane and placed at 4 °C overnight, allowing for metabolite transfer. Nitrocellulose membranes were then removed from plates and soaked in Salkoski reagent (0.01 M ferric chloride in 35% perchloric acid) for 30 min. Reddish pink halos were visualized on an illuminated background and indicate bacterial IAA production [36].

### 4.7. Seed-Borne Fungi from Cannabis Culturing

Five seeds of each *Cannabis* accession were planted on PDA + 20 μg/mL tetracycline (inhibits bacterial growth), in a biosafety cabinet and incubated for 5 days at 25 °C to allow for germination and/or growth of fungal colonies. Morphologically unique colonies of fungi associated with seeds and seedlings were subcultured and isolated on PDA. Seed germination and number of seeds/seedlings engulfed in fungal colonies were recorded. Fungal isolates were then subject to colony PCR with primers ITS86F and ITS4R using the following settings: initial denaturation at 95 °C for 5 min, followed by 45 cycles of denaturation at 95 °C for 30 s, annealing at 50 °C for 30 s and extension at 72 °C for 1 min; a final extension phase was performed at 72 °C for 10 min. Amplicons were run on a 1% TAE gel to confirm successful amplification prior to sending to BioBasic (Markham) for Sangar Sequencing. Taxonomy was determined by BLAST against the ITS ribosomal RNA sequences database; highest identity alignments were assigned as isolate taxonomy. Amplicon sequences were submitted to Genbank and assigned accession numbers OP18318-OP183206.

### 4.8. Anti-Fungal Activity of Cannabis Endophytes

Unique bacterial endophytes from each Cannabis accession were selected to assay for anti-fungal activity against seed-borne Cannabis pathogens: *Alternaria* PM35 from Altair, *Aspergillus* PM40 from CAN3797, *Penicillium* PM52 from God’s Bud, and *Fusarium* PM60 from VIR577 (Appendix A). Modifying an established assay [78], bacterial endophytes were streaked on R2A plates from glycerol stocks and grown for 2 days at 25 °C. Single colonies where then used to inoculate TSB liquid medium which was grown at 20 °C for 2 days shaking at 150 RPM.

Fungal cultures were grown on PDA for 5 days prior to harvesting spores by pipetting 1.3 mL sterile 0.1% Tween20 on plates and harvesting by gentle scraping with an L-spreader. An amount of 1 mL of this solution was used to inoculate 2 L of molten PDA cooled to 50 °C, which was poured into 150 mm × 15 mm plates to solidify. A sterilized Pasteur pipette was used to bore 14 holes in each agar plate. An amount of 100 μL of bacteria liquid culture was pipetted into holes into the agar plate. All 78 isolates were tested in triplicate across different agar plates. Uninoculated TSB was used as a control in this experiment. Plates were incubated for 7 days at 25 °C to allow fungi to grow, after which zones of inhibition surrounding bacterial cultures were measured and recorded. Zones with uneven diameters were measured using the shortest distance that bisected the agar hole. Statistical analysis was done using ANOVA and Tukey’s HSD test on RStudio (2021.09.2 + 382). Means not statistically different from each other are denoted by the same letter groups.

## Figures and Tables

**Figure 1 plants-11-02127-f001:**
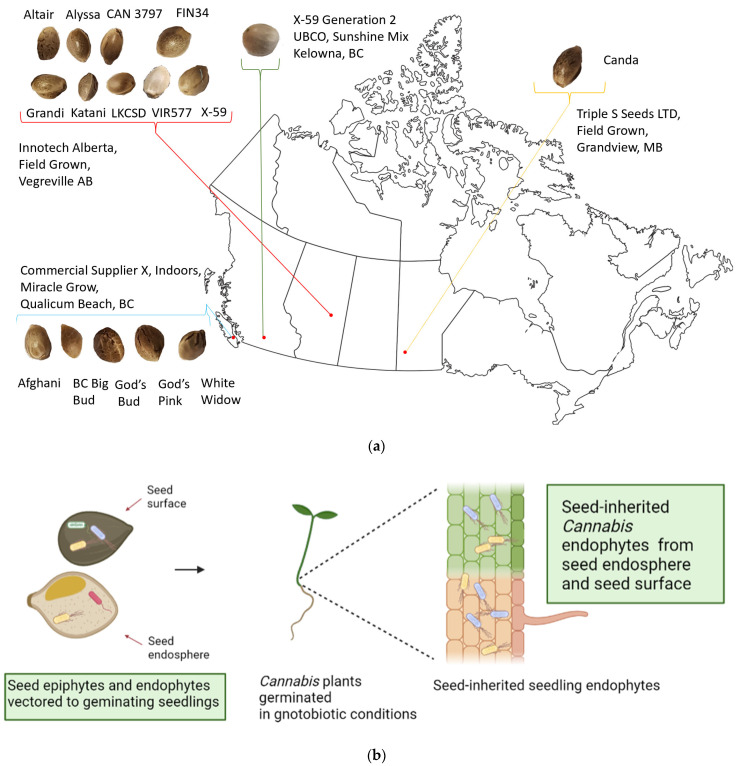
(**a**) Geographic origin of parental *Cannabis sativa* accessions used for seed in this study. All experiments were performed at UBCO (Kelowna, BC, Canada). (**b**) Schematic for the location and isolation of seedling endophytes in this study. Both seed surface and seed endosphere contribute to seed-borne microbiome. Figure 1b was created with BioRender.com.

**Figure 2 plants-11-02127-f002:**
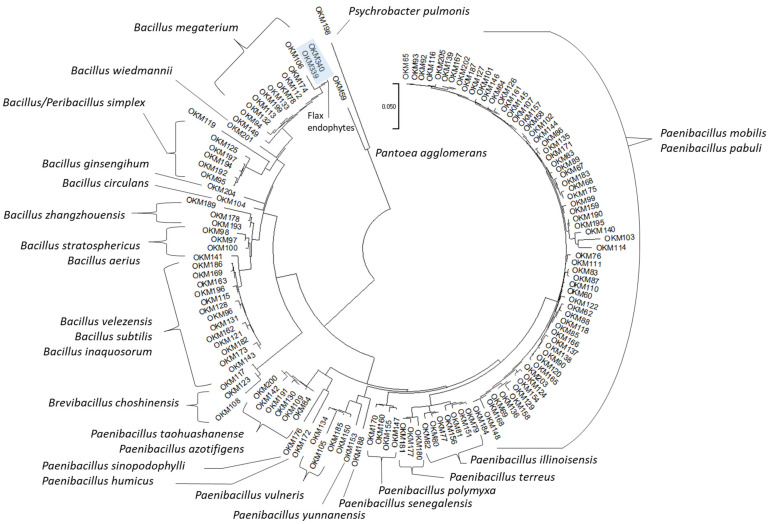
Nearest-Neighbor Phylogenetic Tree of 16S Amplicons from Culturable Seed-Borne Bacterial Endophytes. Amplicons were aligned using the MUSCLE algorithm prior to nearest-neighbor phylogenetic tree construction using the Tamura–Nei model on MEGA software (Version 11.0.11).

**Figure 3 plants-11-02127-f003:**
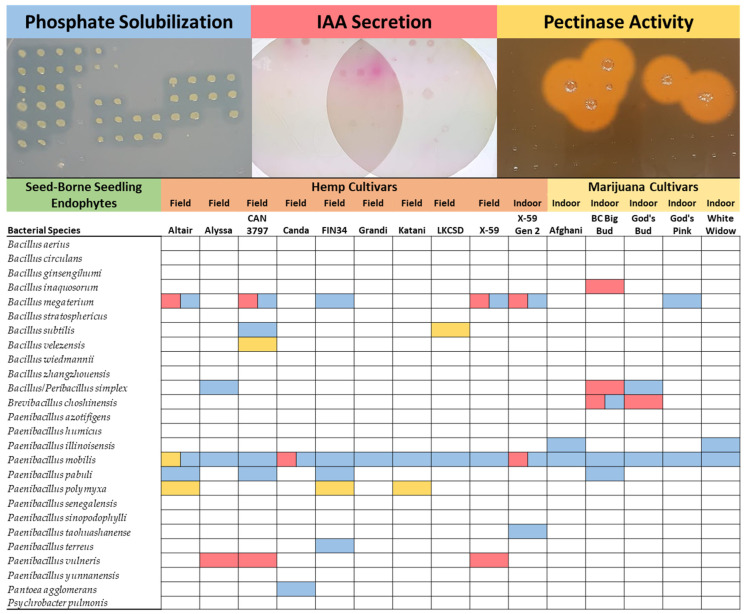
Summary of in vitro Phenotyping Assays. *Cannabis* endophytes were assayed for the potential to solubilize mineral phosphate, secrete IAA, and hydrolyze pectin. Blue squares indicate phosphate solubilization. Red squares indicate IAA secretion. Yellow squares indicate pectinase activity.

**Figure 4 plants-11-02127-f004:**
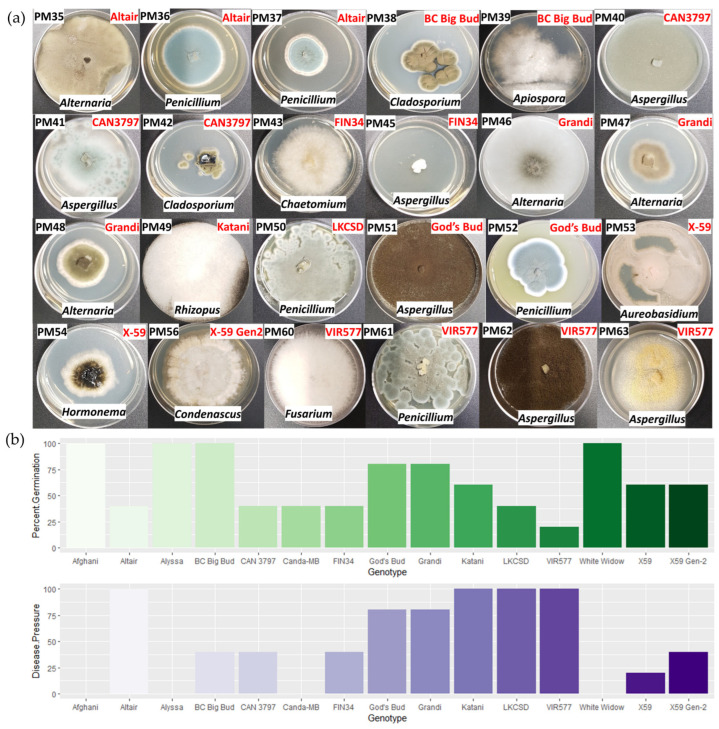
Seed-Borne Fungi from *Cannabis*. (**a**) Fungal isolates from seeds of 15 *Cannabis* accessions germinated on PDA for 5 days. (**b**) Germination rates and percent of seeds with visible mycelia growth (disease pressure) for each *Cannabis* accession.

**Figure 5 plants-11-02127-f005:**
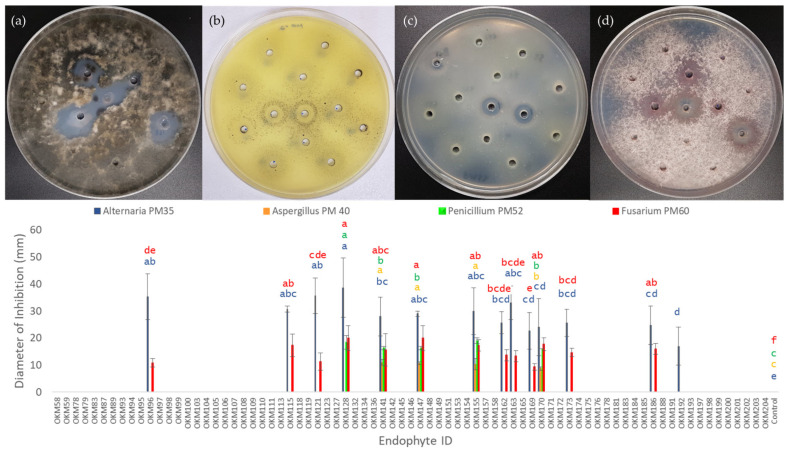
Biocontrol Activity of 78 Seed-Borne Cannabis Endophytes Against Seed-Borne Fungi. Four fungal isolates belonging to plant pathogenic genera were selected (**a**) Alternaria, (**b**) Aspergillus, (**c**) Penicillium, (**d**) Fusarium. Diameters of zones of inhibition were measured at 7 days post-inoculation with endophytes. Color-coded letters correspond to means within the same group as calculated by ANOVA and Tukey HSD test.

**Table 1 plants-11-02127-t001:** Culturable Seed-borne Endophytes from Seedling Endospheres. Green shaded cells indicate isolation from pool of 5 seedlings following germination in sterile environment and surface sterilization.

Seed-Borne Seedling Endophytes	Hemp Cultivars										Marijuana Cultivars					Flax
	Field	Field	Field	Field	Field	Field	Field	Field	Field	Indoor	Indoor	Indoor	Indoor	Indoor	Indoor	Field
Bacterial Species	Altair	Alyssa	CAN 3797	Canda	FIN34	Grandi	Katani	LKCSD	X-59	X-59 Gen 2	Afghani	BC Big Bud	God’s Bud	God’s Pink	White Widow	Bethune
*Bacillus aerius*																
*Bacillus circulans*																
*Bacillus ginsengihumi*																
*Bacillus inaquosorum*																
*Bacillus megaterium*																
*Bacillus stratosphericus*																
*Bacillus subtilis*																
*Bacillus velezensis*																
*Bacillus wiedmannii*																
*Bacillus zhangzhouensis*																
*Bacillus simplex*																
*Brevibacillus choshinensis*																
*Paenibacillus azotifigens*																
*Paenibacillus humicus*																
*Paenibacillus illinoisensis*																
*Paenibacillus mobilis*																
*Paenibacillus pabuli*																
*Paenibacillus polymyxa*																
*Paenibacillus senegalensis*																
*Paenibacillus sinopodophylli*																
*Paenibacillus taohuashanense*																
*Paenibacillus terreus*																
*Paenibacillus vulneris*																
*Paenibacillus yunnanensis*																
*Pantoea agglomerans*																
*Psychrobacter pulmonis*																

**Table 2 plants-11-02127-t002:** Summary of Anti-fungal Activity in Seed-borne Cannabis Endophytes.

Strain ID	Taxonomy	Accession	Host	Antagonizes
OKM96	*Bacillus inaquosorum*	BC Big Bud	Marijuana	*Alternaria*
OKM115	*Bacillus subtilis*	BC Big Bud	Marijuana	*Alternaria, Fusarium*
OKM121	*Bacillus subtilis*	God’s Bud	Marijuana	*Alternaria, Fusarium*
OKM128	*Bacillus subtilis*	CAN 3797	Hemp	*Alternaria, Fusarium, Penicillium*
OKM141	*Bacillus velezensis*	CAN 3797	Hemp	*Alternaria, Aspergillus, Fusarium, Penicillium*
OKM147	*Paenibacillus polymyxa*	Altair	Hemp	*Alternaria, Aspergillus, Fusarium, Penicillium*
OKM155	*Paenibacillus polymyxa*	Katani	Hemp	*Alternaria, Aspergillus, Fusarium, Penicillium*
OKM162	*Bacillus subtilis*	Grandi	Hemp	*Alternaria, Fusarium*
OKM163	*Bacillus subtilis*	Altair	Hemp	*Alternaria, Fusarium*
OKM169	*Bacillus subtilis*	LKCSD	Hemp	*Alternaria, Fusarium*
OKM170	*Paenibacillus polymyxa*	FIN34	Hemp	*Alternaria, Aspergillus, Fusarium, Penicillium*
OKM173	*Bacillus subtilis*	FIN34	Hemp	*Alternaria, Fusarium*
OKM186	*Bacillus subtilis*	Alyssa	Hemp	*Alternaria, Fusarium*

## Data Availability

Sequences in this study were submitted to Genbank under accession numbers OP087663-OP087800 and OP183181-OP183206.

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
