# Peer review of "Cannabis Seedlings Inherit Seed-Borne Bioactive and Anti-Fungal Endophytic Bacilli"

_plants, 2022, doi:10.3390/plants11162127_

Round 1

Reviewer 1 Report

This is an interesting study on “Cannabis Seedlings Inherit Seed-Borne Bioactive and 2

Antifungal Endophytic Bacilli” and the authors have collected a unique dataset using the cutting-edge methodology. The paper is generally well written and structured. However, in my opinion, the article has some shortcomings regarding data analyses and text, and I feel this unique dataset has not been utilized to its full extent. Below I have provided numerous remarks on the text as it is often vague and long-winded. In several instances, I also suggested citing more relevant and recent literature. Furthermore, I have made line by additional line suggestions for more in-depth analyses of the data. Key critical points are:-

Title: Please change the title to “Cannabis Seedlings Inherit Seed-Borne Bioactive and 2

Antifungal Endophytic Bacilli: A potential source for biological mediated management.   

Abstract

Please follow the below-mentioned comments to improve the quality of the paper:-

Line no.12: Please correct with an adaptation that plants can use to protect……..

 Line No 18: Mentioned fungal genera which Cannabis seeds carry are from soil-borne and air-borne sectors. Can you specify the presence of these fungi which are the pathogens to Cannabis, as fusarium cause wilt/ damping off, Alternaria cause leaf spots, and the rest are vagabond fungus genera prevailing everywhere

Line 20: Please add 2 to 3 sentences from your results with exclusive emphasis on the bioactive fungus and bacteria

Line 22-23: As this is a new and unique study so you should mention future avenues for research on this domain in one to two sentences

Keywords

Line No. 24-25, Remove the words mentioned in the title, and please add biological mediated management in keywords.

Introduction

Line No 34-35- Please rephrase and merge both sentences of lines 34 and 35.

Line 37. Please add a reference after this sentence.

Line 40. Only the word microbiome is sufficient to mention here and also specify. Are you describing the microbiome of seeds?

Line 48. Specify the seeds microbiome and rhizosphere micro-biome and their interaction in this sentence in context to roots and rhizosphere

Line 69. Add reference here at the end of a sentence

At the end of the introduction section, please mention your objectives separately with numerics

Results

Results are well written

Line 154 -155 Rewrite this sentence

Line 192- 193. Please merge both sentences to make them more impactful and meaningful

Discussion

The discussion has been excellent with all the support and vitae of reference.

I suggest strengthening your study with the latest references from the last four years of publications in this domain.

Materials and Methods

Line 431. 4.2. Support your methodology with the latest reference at the end of the paragraph

Line 457. 4.4. Support this methodology with the appropriate reference

Line 481. 4.7. Support the isolation of fungi with the latest reference at the end of the paragraph

Line 495. 4.8. I will suggest mentioning Biological mediated antifungal activity of cannabis endophytes

Author Response

Thank you for your comments to improve our manuscript. Please see the attached document outlining point by point responses your your comments and see the revised manuscript for their integration 

Reviewer 2 Report

The authors report an interesting manuscript on the competitive interaction between bacteria and fungi that colonize the seeds of two cultivars of Cannabis sativa.

The topic addressed is very important for the agronomic sciences. In particular, understanding the intricate interaction between seed microbiome, growth hormones, root and soil microbiome, is essential to create crops that are resistant to the stresses associated with different climatic conditions.

The authors modeled a cannabis sativa with high THC levels (marijuana cultivar) and another cultivar with low THC levels (hemp cultivar). They also analyzed the main microbial and fungal components present during germination and growth. Furthermore, they evaluated the variation in auxin levels during all phases studied. Finally, they established an "evolutionary-competitive" interaction between the bacteria and fungi that colonize Cannabis sativa.

The manuscript is clear and many findings have been presented and discussed.

Given the importance of the topic, and how it was argued by the authors, the manuscript can be published.

Author Response

Thank your for taking the time to review our paper and for your support of our manuscript.
